# High temperature Néel skyrmions in simple ferromagnets

Peng Wang[1], Rana Saha [1,2], Holger L. Meyerheim [1], Ke Gu [1], Hakan Deniz[1], David Eilmsteiner [3], Andrea Migliorini [1], Banabir Pal [1], Juan Rubio Zuazo[4,5], Eugenia Sebastiani-Tofano[4,5], Ilya Kostanovski [1], Abhay Kant Srivastava[1], Arthur Ernst [1,3] & Stuart S. P. Parkin [1] ✉

A wide variety of chiral non-collinear spin textures have been discovered and have unique properties that make them highly interesting for technological applications. However, many of these are found in complex materials and only in a narrow window of temperature. Here, we show the formation of Néel-type skyrmions in thin layers of simple ferromagnetic alloys, namely Co-Al and Co-Ni-Al, over a wide range of temperature up to ~773 K, by imposing a strain gradient perpendicular to the sample plane via epitaxy with an Ir-Al under-layer. The Néel skyrmions are directly observed using Lorentz transmission electron microscopy in freestanding membranes at high temperatures and the strain gradient is directly measured from x-ray diffraction asymmetric peak profiles. Our concept allows for simple centrosymmetric ferromagnets with high magnetic ordering temperatures to exhibit skyrmions at temperatures well above room temperature, thereby, bringing closer skyrmionic electronics.

Recently, a zoology of topological chiral spin textures has been discovered in a wide range of ferro- and ferri-magnetic compounds, typically stabilized within a narrow temperature window[1,2]. These spin textures include Bloch[3] and Néel[4,5] skyrmions, perhaps the simplest of these spin textures, as well as more complex textures such as anti-skyrmions[6] and many more[7,8]. Each of these is typically nanoscopic in size, ranging from tens to hundreds of nanometers[9,10], and exhibits many shapes and forms[7,11]. Beyond their fundamental interest, the application of skyrmions as non-volatile bits for non-volatile memory and unconventional computing devices has been proposed[12–14]. One of the biggest challenges to date is the stabilization of skyrmions at temperatures well above ambient temperature, especially in thin films.

A prerequisite for the formation of many of the skyrmionic spin textures is the presence of a vector Dzyaloshinskii–Moriya magnetic exchange interaction (DMI)[15,16] in addition to the dominant Heisenberg exchange. A DMI is possible in bulk magnetic compounds that lack crystal inversion symmetry[17,18]. Indeed, one avenue to search for novel skyrmionic spin textures has been to explore magnetic materials with a certain crystal symmetry that is then reflected in the form of the DMI.

This led, for example, to the discovery of the anti-skyrmion in two distinct materials[6,19]. Another strategy is to form hetero-interfaces between thin magnetic layers and non-magnetic (or anti-ferromagnetic) layers that typically contain heavy atoms, which thereby gives rise to an interface-derived DMI[20]. These limitations considerably restrict the number of suitable magnetic materials that exhibit skyrmionic spin textures. An alternative strategy to introduce a DMI in thin films is to remove the inversion symmetry of a high symmetry material, for example, by the introduction of a strain gradient ($\nabla_t \varepsilon$), such as by mechanical means[21], by ripples in thin film membranes[22], by composition variation[23], or by thin film epitaxy[24–26]. Perhaps the latter is the most elegant. Indeed, strain gradients normal to thin layers have been induced in several magnetic oxide thin film systems by epitaxial growth[24–26], but it has been proven to be much more difficult to establish a strain gradient in metallic films[27,28]. Here, we show the formation of large vertical strain gradients over considerable thicknesses in metallic films formed from ferromagnetic cubic Co–Al and Co–Ni–Al alloys, which have very high Curie temperatures, by depositing them on a very special underlayer formed

[1]Max Planck Institute of Microstructure Physics, Halle (Saale), Germany. [2]Department of Chemistry, Indian Institute of Science Education and Research, Tirupati, India. [3]Institute for Theoretical Physics, Johannes Kepler University Linz, Linz, Austria. [4]Spanish CRG Beamline BM25-SpLine at the ESRF, Grenoble, France. [5]Instituto de Ciencia de Materiales de Madrid-CSIC, Madrid, Spain. ✉e-mail: stuart.parkin@mpi-halle.mpg.de

from the $L1_0$ ordered alloy Ir−Al. Furthermore, we show that the vertical strain gradient in these layers gives rise to a large DMI via the direct observation of Néel-type skyrmions through Lorentz transmission electron microscopy. The skyrmions exist up to very high temperatures, as high as ~773 K, that are higher than the highest temperature at which skyrmions have previously been observed in any material. Thus, we introduce a novel strategy for creating skyrmions at very high temperatures in simple ferromagnetic layers by the introduction, via thin film epitaxy, of large strain gradients.

## Results

Typically, thin films deposited on a substrate, via single-crystalline or polycrystalline epitaxy, maintain a constant strain ($\varepsilon$) up to a critical thickness at which the strain is released. We show that thin films of the tetragonal ferromagnets $Co_{2.3}Al$ and $Co_{2.58}Ni_{0.26}Al$ rather display a significant strain gradient perpendicular to the sample plane when deposited on a suitable underlayer and within an optimal thickness range of 30 to 50 nm. To achieve large strain gradients, we explored engineered underlayers composed of the M-Al alloys (M = Ir, Pd, and Ru). These cubic alloys were chosen because at ambient temperature they grow as flat films with a highly chemically ordered $L1_0$-type crystal structure[29,30] and are characterized by a larger lattice parameter than the ferromagnetic layers considered here. This lattice mismatch plays a key role in generating the strain gradient. However, of these alloys only IrAl was found to give rise to a significant strain gradient in the magnetic overlayers. The largest strain gradient ($\nabla_t\varepsilon = 3.8 \times 10^{-3}$/nm) was found in ferromagnetic layers of $Co_{2.58}Ni_{0.26}Al$. For strain gradients larger than approximately $7 \times 10^{-4}$/nm, Néel skyrmions are observed by Lorentz Transmission Electron Microscopy (LTEM) in both $Co_{2.3}Al$ and $Co_{2.58}Ni_{0.26}Al$ layers prepared on IrAl underlayers. Ab initio

calculations establish that the largest strain gradients found experimentally give rise to substantial values of DMI that can account for the Néel skyrmions we observe both via LTEM measurements and by micromagnetic simulations. For the case of $Co_{2.3}Al$, high-temperature LTEM studies using films prepared in the form of freestanding membranes show clear evidence for skyrmions up to very high temperatures of ~773 K.

The strain gradients in the epitaxially grown films were determined by fitting X-ray diffraction (XRD) reflection profiles collected by line scans along the out-of-plane ($q_z$) direction in the vicinity of the (002) reflection in reciprocal space. The simulation of the reflection profiles was carried out by employing the recursive matrix formalism in the kinematic scattering approximation[31,32]. The film is subdivided into thin slices, each having a different vertical strain along the surface normal [001], as schematically shown in Fig. 1a, b. Here, and in the following, we choose to refer to the strain to the out-of-plane $c$-lattice parameter of the cubic IrAl underlayer (2.985 Å). Additional fitting parameters include the film thickness $t$ and its root-mean-square (rms) surface roughness ($\sigma$) modeled by a Debye−Waller-type approach. A least squares fit of the simulated curve to the experimental data was carried out using a standard $\chi^2$ minimization procedure[33]. Results for $Co_{2.3}Al$ layers with thicknesses ranging from 20 to 70 nm and a 27 nm thick $Co_{2.58}Ni_{0.26}Al$ layer, each grown on top of 4.3 nm IrAl are shown in Fig. 1c. Further results for $Co_{2.3}Al$ layers grown without and with IrAl underlayers are presented in Supplementary Figs. S1−S6. The experimental (002) reflection profiles (symbols) are shown together with the corresponding calculated ones (solid line) on a logarithmic intensity scale. The analysis indicates that these films exhibit a significant positive strain gradient, which ranges from ~$8.2 \times 10^{-4}$/nm up to $3.8 \times 10^{-3}$/nm. As an example shown in Fig. 1b, we consider the refined

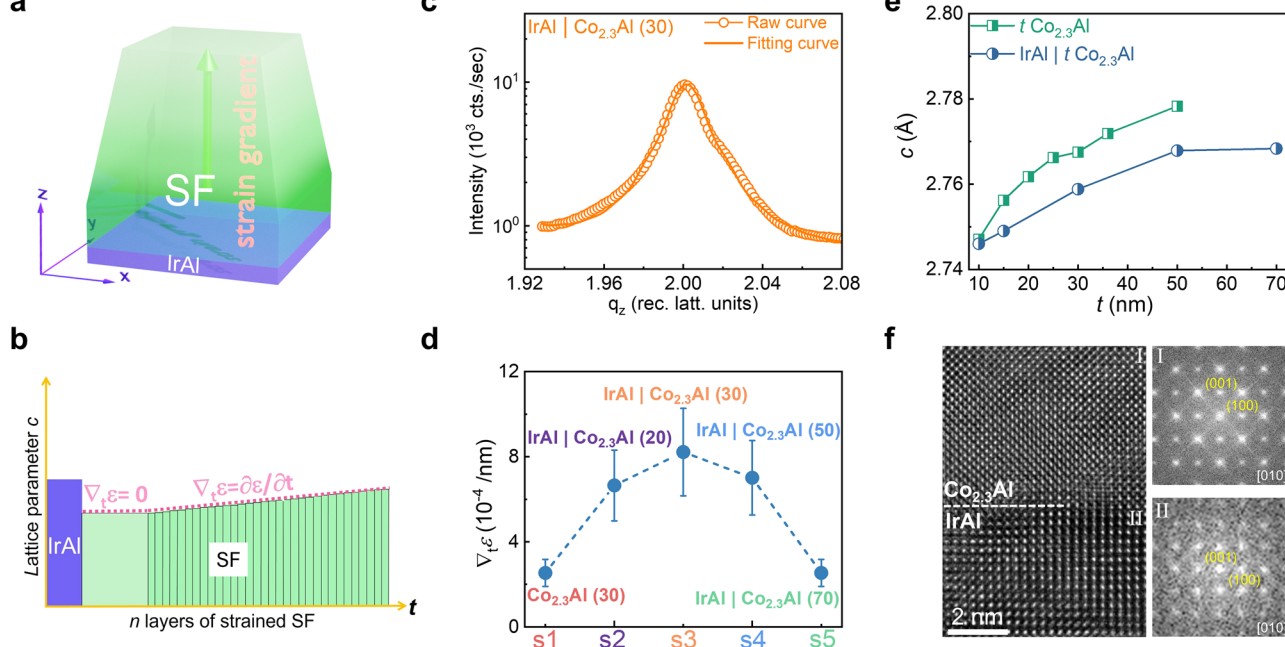

**Fig. 1 | Strain gradient analysis and structural properties. a** Schematic illustration of the epitaxial relationship between the strain gradient induced in a ferromagnetic film (SF) via an IrAl(001) underlayer. **b** Schematic of the XRD-derived strain profile of the 50 nm thick $Co_{2.3}Al$ film on IrAl(001) (s4). A constantly strained layer ($\nabla_t\varepsilon = 0$) is located between the IrAl layer and the SF, which is characterized by a linear positive strain gradient. **c** Exemplary X-ray diffraction profile on a log scale in the vicinity of the L = 2 reflection along the [00 L] direction in reciprocal space for sample s3 that has a structure of the form 4.3 nm IrAl | 30 nm $Co_{2.3}Al$. Experimental data are shown as circles, and the solid line is a fit to the data. The pronounced asymmetry in the peak profile indicates a large strain gradient, $\nabla_t\varepsilon = 8.2 \times 10^{-4}$/nm.

**d** Thickness-dependent strain gradient at ambient temperature for 5 samples: s1: 30 nm $Co_{2.3}Al$; s2: 4.3 nm IrAl | 20 nm $Co_{2.3}Al$; s3: 4.3 nm IrAl | 30 nm $Co_{2.3}Al$; s4: 4.3 nm IrAl | 50 nm $Co_{2.3}Al$; s5: 4.3 nm IrAl | 70 nm $Co_{2.3}Al$. The error bars correspond to the standard deviation in the fits of the XRD profiles. **e** Comparison of the thickness dependence of the average $c$-lattice parameter of $Co_{2.3}Al$ layers grown directly on MgO (001) single crystalline substrates and with 4.3 nm thick IrAl underlayers. **f** Cross-sectional, bright field, high-resolution transmission electron microscopy (HR-TEM) image of s3. Corresponding Fourier transforms (FT) of two different regions (I and II) show the epitaxial relationship between the IrAl and $Co_{2.3}Al$ layers.

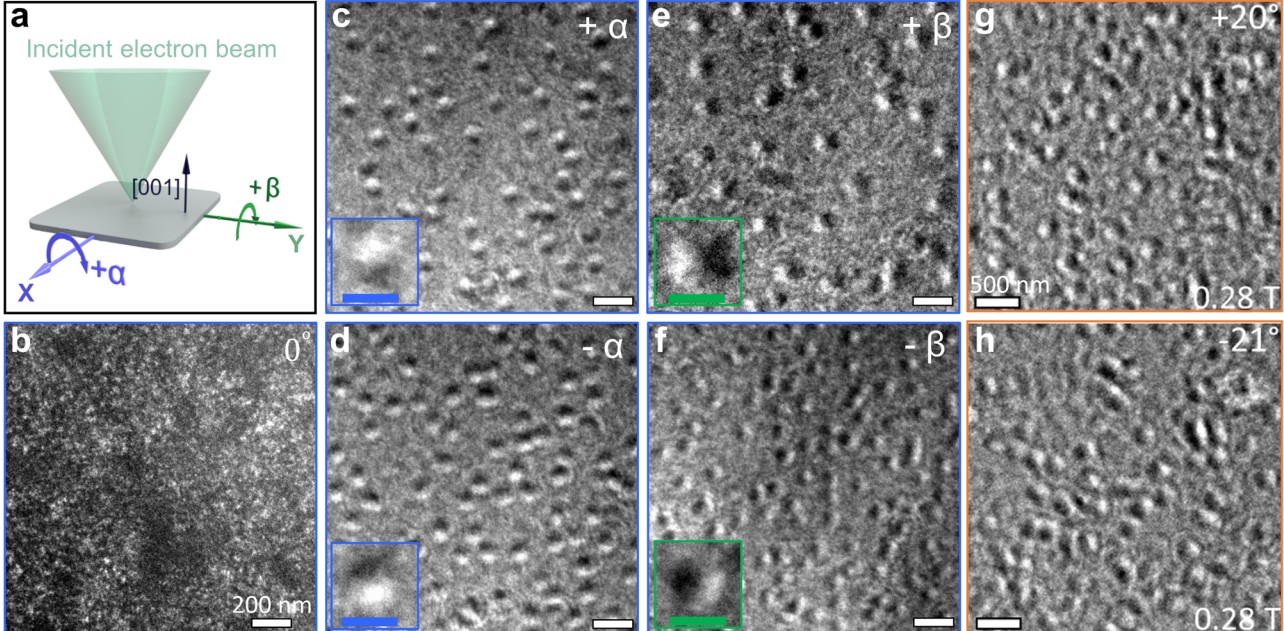

**Fig. 2 | Lorentz transmission electron microscopy (LTEM) images of free-standing films at 300 K. a** Schematic illustration of specimen tilting inside the electron microscope. Zero tilt corresponds to the incident electron beam being oriented along the surface normal [001] of the lamella. **b–f** LTEM images recorded for a 4.3 nm IrAl | 30 nm $Co_{2.3}Al$ bilayer in the presence of a 0.25 T magnetic field applied along the TEM column. **b** LTEM image at zero tilt shows no magnetic contrast. **c–f** LTEM images recorded upon sample tilting about the $x$ axis (**c, d**) $\alpha = \pm30°$, and the $y$ axis (**e, f**), $\beta = \pm30°$. The insets in (**c–f**) show magnified images of the highlighted (blue or green circle) nano-objects that correspond to Néel skyrmions. All images are taken at a defocus value of −1.5 mm. The scale bar for (**b–f**) is 200 nm, and the scale bar inset (**c–f**) is 100 nm. **g, h** LTEM images of bilayer of 4.3 nm IrAl | 27 nm $Co_{2.58}Ni_{0.26}Al$ in the presence of a 0.28 T magnetic field applied along the TEM column at 300 K. Defocus was set to −0.6 mm, and the tilt angle, $\alpha$, is set to +20° and −21°, as shown in the figure. The scale bar for (**g, h**) is 500 nm.

strain profile within the 50 nm thick $Co_{2.3}Al$ film. Directly above the IrAl layer, we find an approximately 9 nm thick $Co_{2.3}Al$ layer, which is constantly strained by about -8.0% ($c = 2.74$ Å) relative to the $c$ lattice parameter of IrAl of 2.985 Å. This layer is followed by an approximately 30 nm thick $Co_{2.3}Al$ layer in which the strain linearly diminishes from -8.0% at the bottom to −5.7% (2.81 Å) at the top, corresponding to a vertical strain gradient of $8.2 \times 10^{-4}$/nm. We note that our XRD analysis revealed no significant changes in the in-plane lattice constants, $a$ and $b$, with varying $Co_{2.3}Al$ thickness for the same IrAl underlayer (and MgO buffer layer) across different samples, nor any significant variation in the $a$ and $b$ values within a given sample across the thickness of the $Co_{2.3}Al$ layer.

The accurate modeling of the strain profile, as outlined above, is a prerequisite to obtain a high degree of agreement between the simulated and experimental reflection curves. A large strain gradient is directly evidenced by the pronounced asymmetry of the X-ray reflection profile, which is well reproduced by the simulations (Fig. 1c and Supplementary Fig. S5). By contrast, a strain gradient in the lower $1.2 \times 10^{-4}$/nm regime results in a nearly symmetric profile as found for thick $Co_{2.3}Al$ layers (Supplementary Fig. S5). The key role of the IrAl underlayer is evidenced by the lack of any strain gradient in a 30 nm $Co_{2.3}Al$ layer prepared without the underlayer (which displays a symmetric XRD profile, see Supplementary Fig. S5), yet which shows a very large strain gradient when prepared with the IrAl underlayer.

In agreement with the (linear) positive strain gradient derived from the profile fitting, we find that the average $c$ lattice parameter of the $Co_{2.3}Al$ films derived by the peak position increases with their film thickness (Fig. 1e and Supplementary Fig. S4). The high quality of the epitaxially grown $Co_{2.3}Al$ films on IrAl was directly demonstrated by cross-sectional high-resolution transmission electron microscopy (HRTEM). A typical image is shown in Fig. 1f, together with Fourier transforms (FT) showing bright, well-defined spots.

Vibrating sample magnetometry (VSM) measurements confirmed an out-of-plane easy axis magnetization of the IrAl | $Co_{2.3}Al$ bilayers for hysteresis loops recorded in the temperature range from 100 to 600 K: exemplary data for two samples are shown in Supplementary Figs. S7–S9. The magnetization, $M$, gradually decreases with increasing temperature with an estimated Curie temperature ($T_c$) of ~1200 K from fitting $M$ (T) to Bloch's law (inset of Supplementary Fig. S7 and Supplementary Information).

Figure 2a and Supplementary Figs. S10 and S11 show schematics of the LTEM configuration in which a thin lamella, prepared by conventional mechanical polishing from a deposited film, is oriented in a double-tilt TEM sample holder so that its surface normal is parallel to the incident electron beam. Thus, this situation corresponds to tilt angles $\alpha = \beta = 0°$, where these angles correspond to rotations about two mutually orthogonal rotation axes lying in the lamella plane. First, we discuss results for a 4.3 nm IrAl | 30 nm $Co_{2.3}Al$ bilayer. No magnetic contrast is observed for $\alpha = \beta = 0°$ (see Fig. 2b and Supplementary Fig. S12a). However, a magnetic contrast appears if the sample is tilted, and many circular objects are seen (see Fig. 2c–f and Supplementary Figs. S12 and S13). Each of these objects exhibits bright and dark regions at its opposite edges. These regions appear along the $x(y)$-axis when the lamella is tilted by $\pm \alpha(\beta)$ (Fig. 2c–f). This magnetic contrast is characteristic of a Néel-type skyrmion[34,35]. We find that these nano-objects are stable over a wide range of magnetic field at ambient temperature and to low temperatures (see Supplementary Fig. S14). In addition to the observation of Néel skyrmions in $Co_{2.3}Al$, clear evidence of Néel-type skyrmions is also provided by LTEM for a 4.3 nm IrAl | 27 nm $Co_{2.58}Ni_{0.26}Al$ thick bilayer thin film, as shown in Fig. 2g, h (for more information, refer to Supplementary Figs. S15 and S16). Note that, for this film, we find the largest strain gradient, $\nabla_t \varepsilon$ ~ $3.8 \times 10^{-3}$/nm, observed in any of the magnetic films studied here.

Next, we discuss in detail five $Co_{2.3}Al$ samples with varying thicknesses of the ferromagnetic layer (see Fig. 3 and Supplementary

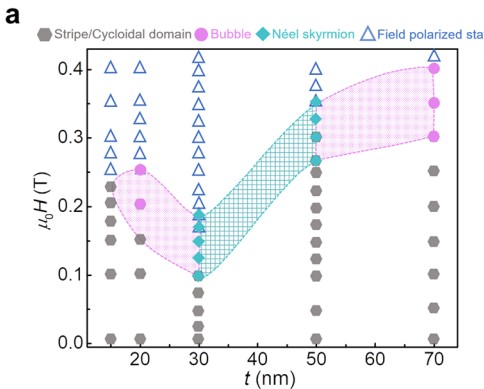

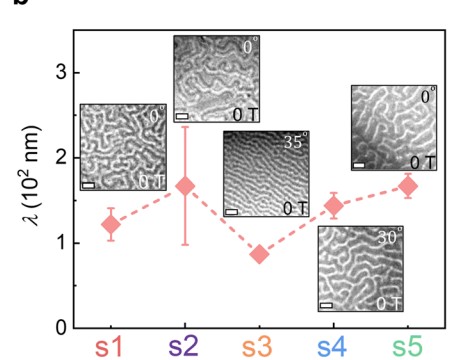

**Fig. 3 | Thickness-dependent magnetic textures at ambient temperature. a** Dependence of the spin texture type derived from LTEM imaging for 4.3 nm IrAl | $t$ $Co_{2.3}Al$ bilayers as a function of the $Co_{2.3}Al$ thickness and applied magnetic field. Different spin textures are labeled by: stripe/cycloidal: 🔲, bubbles: 🟣, Néel skyrmion: 🔷, and field polarized state: △. **b** Variation of period, $\lambda$, of stripe/ cycloidal phase versus sample number for five samples: s1: 30 nm $Co_{2.3}Al$; s2: 4.3 nm IrAl | 20 nm $Co_{2.3}Al$; s3: 4.3 nm IrAl | 30 nm $Co_{2.3}Al$; s4: 4.3 nm IrAl | 50 nm $Co_{2.3}Al$; s5: 4.3 nm IrAl | 70 nm $Co_{2.3}Al$. The insets show the corresponding LTEM images of stripe/cycloidal domains in zero magnetic field, and the scale bar in each LTEM image is 500 nm. The error bars represent the standard deviation in $\lambda$.

Figs. S18–S23). A detailed field ($\mu_0 H$) versus thickness ($t$) phase diagram showing the stability regions of the different spin textures in these samples is outlined in Fig. 3a. Samples s1 and s5 exhibit X-ray reflection profiles that are nearly symmetric (see Supplementary Fig. S5) corresponding to $\nabla_t \varepsilon$ lying in the low $2.4 \times 10^{-4}$/nm regime (Fig. 1d). For these samples, Néel-type skyrmions cannot be stabilized, and only stripe domains and/or type-II bubbles are observed (Supplementary Fig. S23). By contrast, samples s3 and s4 exhibit the largest strain-gradients (up to $8.2 \times 10^{-4}$/nm) among these five samples (see Fig. 1d) and Néel skyrmions are observed (Supplementary Fig. S23). We find, therefore, that Néel skyrmions are only observed for film thicknesses in the 30 to 50 nm regime and for intermediate magnetic field strengths, as shown in Fig. 3a. Thinner films exhibit stripe-type domains, whereas thicker films exhibit both stripe domains[36] and type-II bubbles[37] (see Supplementary Fig. S23). Apart from the formation of Néel skyrmions, the effect of the strain gradient is also clearly manifested in the cycloidal domains observed in the absence of a magnetic field, as discussed in Fig. 3b. Here, the size ($\lambda$) of the cycloidal domains displays a clear inverse relationship with the strain gradient ($\nabla_t \varepsilon$) (except for a large uncertainty in sample s2). The distinction between the cycloidal phase and stripe domains can be identified from tilting experiments in LTEM. The cycloidal phase, characterized by Néel-type walls, generates contrast only when the sample is tilted, whereas stripe domains with Bloch walls would produce contrast even without tilting.

The absence of skyrmions in the thinnest $Co_{2.3}Al$ layers may be attributed to the constant strain region near the IrAl underlayer that we identify from the detailed fitting of the XRD peak profile, as mentioned above. Another possibility is the higher effective perpendicular magnetic anisotropy that we measure for the thinner magnetic layers, as shown in Supplementary Fig. S24. It has been shown that there is a critical DMI constant ($D_c$) above which cycloidal domains are formed that increases with $\sqrt{K_{eff}}$[38,39], where $K_{eff}$ is the effective magnetic anisotropy. Thus, $D_c$ is much larger for the thinner layers, which is consistent with the absence of skyrmions.

We find that only type-II bubbles are found in $Co_{2.3}Al$ films without an underlayer, irrespective of thickness (for more information, refer to Supplementary Fig. S25), as shown in Supplementary Figs. S26 and S27. In order to investigate the role of a possible interfacial DMI contribution between IrAl and $Co_{2.3}Al$ in stabilizing Néel-type skyrmions, two sets of samples were prepared that include, in one case, the insertion of a 0.3 nm thick aluminum dusting layer between 4.3 nm IrAl and 30 nm $Co_{2.3}Al$ and, in a second case, the addition of a 2 nm thick IrAl layer on top of the 4.3 nm IrAl | 30 nm $Co_{2.3}Al$ bilayer structure. Néel skyrmions were observed in both cases without any change in their

diameter (see Supplementary Figs. S28 and S29) showing that the role of any interfacial DMI is negligible as compared to the strain gradient effect (also see Supplementary Fig. S30). Furthermore, when the composition ($x$) of $Co_x Al$ is modified to $x = 2.0, 2.6$, and $2.9$ (Supplementary Fig. S31 and S32), or when IrAl is replaced by PdAl or RuAl (Supplementary Fig. S33), only stripe domains and type-II bubbles are found (see Supplementary Figs. S34 and S35). Detailed XRD profile analysis indicates that thin films with PdAl or RuAl underlayers do not produce the necessary strain gradient ($\nabla_t \varepsilon \leq 10^{-4}$/nm) to generate sufficient DMI for stabilizing the stripe domains.

To explore the thermal stability of the Néel skyrmions, in situ LTEM experiments were carried out up to temperatures as high as ~773 K. A novel method was developed to make this experiment possible. A freestanding membrane formed from 12 nm MgO | 4.3 nm IrAl | 30 nm $Co_{2.3}Al$ was prepared by first growing this structure on a sacrificial layer of $Sr_3 Al_2 O_6$ that had been deposited by pulsed laser deposition on a $SrTiO_3(001)$ substrate. The $Sr_3 Al_2 O_6$ layer was then dissolved in water from the edges of the substrate and the membrane floated off[40], and subsequently transferred to a heating chip as shown in Fig. 4a (see "Methods" for more details). The chip was placed on a TEM sample holder that allows for in situ heating[41]. The membrane was then heated in steps of 100–773 K at a heating rate of 5K/s. From selected area electron diffraction (SAED), the crystal structure of the membrane was found to be highly thermally stable (see Supplementary Fig. S36). LTEM images were recorded at each temperature using the same protocol (see Supplementary Figs. S37 and S38). The temperature was increased in a zero field and a zero tilt. After the temperature was stabilized, the sample was tilted (to $\alpha = 20°$), and then the field was increased in steps of 25 mT and a LTEM image taken at each field until the sample was fully field polarized.

Remarkably, we find that the skyrmion phase in $Co_{2.3}Al$ shows exceptional thermal stability, with skyrmions persisting up to ~773 K, the highest temperature reported to date, and well beyond prior records. Previously, skyrmions have been reported to temperatures of up to ~345–360 K in $Fe_3 GaTe_2$ (~360 K)[42–44], Co−Zn−Mn alloys (~ 345 K)[45], and synthetic multilayers, such as Ir|Fe|Co|Pt (~ 350 K)[46]. However, these approaches face limitations in thermal robustness or fabrication complexity. Here, we demonstrate a simple and reliable technique using epitaxial engineering to form high-temperature Néel skyrmions in centrosymmetric materials with high magnetic ordering temperature by introducing a strain gradient. This thermal stability of the skyrmions is not only scientifically significant but is also critical for technological applications, for example, for compatibility with complementary metal-oxide-semiconductor (CMOS) electronics, which

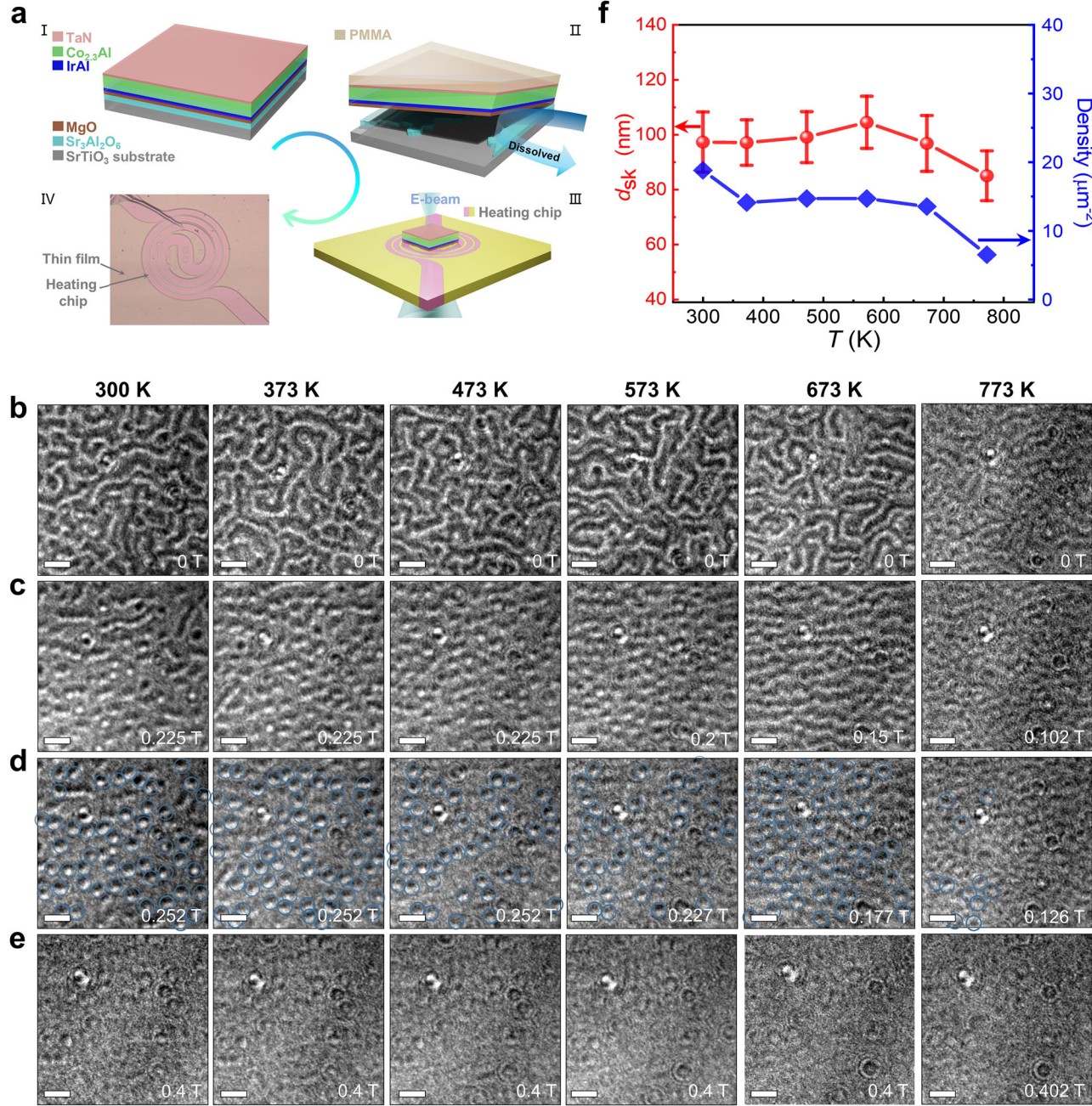

**Fig. 4 | Lorentz transmission electron microscopy (LTEM) images of a free-standing 4.3 nm IrAl | 30 nm Co$_{2.3}$Al bilayer as a function of temperature.**
**a** Schematic diagram of the formation and transfer of a membrane with the structure 12 nm MgO | 4.3 nm IrAl | 30 nm Co$_{2.3}$Al | 5 nm TaN onto a heating chip for in situ LTEM measurements. I. Schematic of a MgO | IrAl | Co$_{2.3}$Al | TaN thin film heterostructure deposited onto a Sr$_3$Al$_2$O$_6$ (SAO) buffer layer grown by pulsed laser deposition on a SrTiO$_3$ (001) substrate. II. Separation of the multilayer film in deionized water by completely dissolving the SAO layer after spin-coating a PMMA (poly(methyl methacrylate)) resist onto the TaN surface. III. Transfer of the thin film onto the heating chip (Lightning, DENS). IV, Optical microscopy image of the

membrane on the heating chip. **b–e** LTEM images recorded at temperatures between 300 and 773 K at selected magnetic fields that are given on each image. LTEM images of **b** cycloidal phase, **c** mixed phase of cycloidal phase and Néel-type skyrmions, **d** Néel-type skyrmions identified with blue circles, and **e** field polarized states. **f** Variation of skyrmion diameter, $d_{sk}$, (red circles) and the skyrmion density (blue squares) as a function of temperature: the error bars represent the standard deviation in the skyrmion diameter. All LTEM images are recorded at a $\alpha = 20°$ tilt at a defocus value of −1 mm. The scale bar in each LTEM image is 200 nm. The error bars represent the standard deviation in $d_{sk}$.

requires operation at temperatures of up to 150 °C. Furthermore, the thermal stability of the material itself makes it compatible with back-end-of-line (BEOL) processing (which requires processing at 400 °C for up to several hours[47,48]). Perhaps the most interesting potential applications are in emerging high-temperature electronics that require even higher temperatures[49]. We note that the electrical control of the formation and manipulation of the skyrmions requires further study.

A transition from a cycloidal domain state into a Néel skyrmion state with increasing magnetic field and, finally, a fully polarized state was found at all temperatures (see Fig. 4b–e). We find that both the skyrmion diameter and density remain constant over a wide temperature range; however, at higher temperatures, both decrease with increasing temperature (Fig. 4f), which is presumably related to the reduction in $K$ at elevated temperatures (see Supplementary Fig. S9).

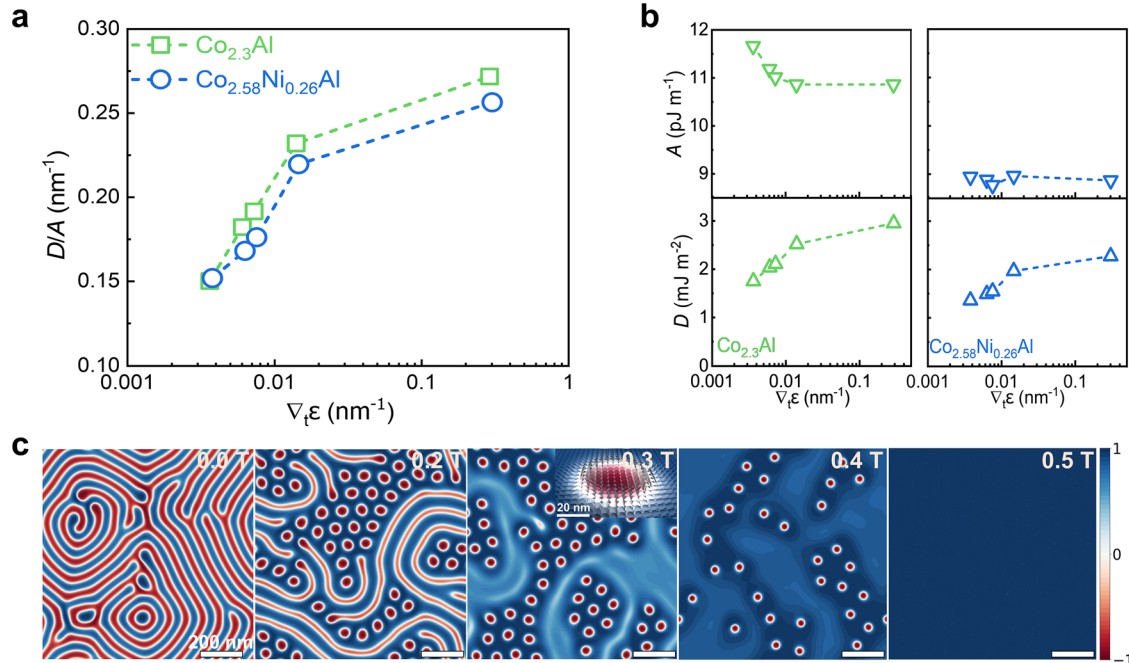

**Fig. 5 | First-principles calculations of the generalized exchange interaction parameters and micromagnetic calculations. a** Ratio $D/A$ versus strain gradient ($\nabla_t \varepsilon$) for $Co_{2.3}Al$ and $Co_{2.58}Ni_{0.26}Al$, **b** $A$ and $D$ versus strain gradient for (left column) $Co_{2.3}Al$ and (right column) $Co_{2.58}Ni_{0.26}Al$. **c** Micromagnetic simulations of the spin textures for $\nabla_t \varepsilon = 3 \times 10^{-3}$/nm as a function of applied magnetic field (see labels) along the surface normal. At $\mu_0 H_z = 0$ T, a cycloidal state is observed. Increasing the magnetic field first leads to a mixture of magnetic stripes and Néel skyrmions, followed by only Néel skyrmions and, finally, a fully polarized magnetic state (from left to right). The inset at $\mu_0 H_z = 0.3$ T shows a close-up view of one Néel skyrmion. The color code depicts the normalized projection of the magnetization onto the surface normal, with +1, 0, and −1 representing up, in plane, and down, respectively. In the simulations, the parameters $D$ and $A$ are taken from first-principles calculations, while the saturation magnetization and the uniaxial magneto-crystalline anisotropy are taken from the current experiments.

The magnetic structures of $Co_{2.3}Al$ and $Co_{2.58}Ni_{0.26}Al$ were further studied using a generalized Heisenberg model,

$$H = -\sum_{ij} J_{ij} \boldsymbol{e}_i \cdot \boldsymbol{e}_j + \sum_{ij} \boldsymbol{D}_{ij} \cdot (\boldsymbol{e}_i \times \boldsymbol{e}_j) + \sum_i K_i (e_i^z)^2 \quad (1)$$

where $\boldsymbol{e}_i$ is the direction of the $i$-th magnetic moment, $e_i^z$ denotes the $z$-component of the unit vector pointing in the direction of the magnetic moment at site $i$, $J_{ij}$ is the exchange interaction between magnetic moments at sites $i$ and $j$, $\boldsymbol{D}_{ij}$ is the Dzyaloshinskii–Moriya vector, and $K_i$ is the microscopic on-site anisotropy tensor[50]. We use first-principles calculations to compute the magnitude of the Heisenberg exchange interaction $J_{ij}$ and the Dzyaloshinskii–Moriya vector $\boldsymbol{D}_{ij}$ as a function of strain gradient. $J_{ij}$ and $\boldsymbol{D}_{ij}$ were obtained using the generalized magnetic force theorem[51–53], as implemented within a multiple scattering theory[54,55]. Magnetic films of various thicknesses were considered that were formed on a semi-infinite IrAl(001) substrate terminated by a semi-infinite CoAl bulk. To avoid strong perturbation effects induced by the IrAl substrate, the spin–orbit interaction in Ir was switched off. Disorder effects were taken into account within a coherent potential approximation[55,56]. A linear strain gradient is considered by assuming an increase of the $c$-lattice parameter beginning with $c_O = 2.7$ Å for the first unit cell (uc) at the $Co_{2.3}Al$ ($Co_{2.58}Ni_{0.26}Al$) interface with IrAl up to $c_n = 2.933$ Å for the $n$-th unit cell at the top surface of the layer. The latter value corresponds to the limit where the topmost unit cell becomes cubic, given the in-plane film lattice parameter of 2.933 Å. Thus, the lattice parameter $c_i$ of the $i$th unit cell from the interface is given by $c_i = c_O + w \times (i-1)$, where $w \approx 0.0826$ nm$^2 \times \nabla_t \varepsilon$, where $\nabla_t \varepsilon$ denotes the strain gradient in /nm (see supplementary information for modeling of the strain gradient). For $\nabla_t \varepsilon = 0.029$/nm, 0.014/nm, 0.007/nm, 0.006/nm, and 0.003/nm, the $Co_{2.3}Al$ and $Co_{2.58}Ni_{0.26}Al$ films consist of 10, 20, 45, 60, and 80 uc's, respectively. The lowest strain gradient corresponds to the largest experimental value found in $Co_{2.58}Ni_{0.26}Al$. Smaller strain gradients were not considered because of the very large computational demand.

We analyze the ratio $D/A$ as a function of strain gradient, where $A$ is the spin stiffness, which corresponds to an effective isotropic exchange parameter, and $D$ is the DMI constant, as conventionally used in micromagnetic simulations. The parameters $A$ and $D$ are defined in detail in Methods and were obtained from the calculated $J_{ij}$ and $D_{ij}$ using a conversion formula outlined in ref. 50. Figure 5a, b shows the calculated $D/A$ ratio as a function of strain gradient in $Co_{2.3}Al$ and $Co_{2.58}Ni_{0.26}Al$, respectively. While the spin stiffness $A$ is almost constant, the DMI parameter $D$ increases monotonically with the strain gradient. For the case of $Co_{2.58}Ni_{0.26}Al$, both $A$ and $D$ are smaller than in $Co_{2.3}Al$, since the presence of Ni reduces the magnetic interactions. Nevertheless, the $D/A$ ratio, which is mainly responsible for skyrmion formation, is calculated to be similar in both systems for similar strain gradients. Experimentally, we find a much larger strain gradient for $Co_{2.58}Ni_{0.26}Al$ as compared to $Co_{2.3}Al$ and, correspondingly, as expected, a larger skyrmion diameter (see Supplementary Figs. S15 and S16).

To model the skyrmion formation, stability, and dynamics, we carried out micromagnetic simulations using the Mumax3 solver[57]. A combination of first-principles calculations and micromagnetic simulations has predictive power and is a well-established approach to study skyrmionic spin textures. Material parameters were partially taken from experiment (saturation magnetization, uniaxial anisotropy, applied magnetic field, and strain gradient) and partially from calculations ($A$, $D$). In agreement with the LTEM images (see Fig. 2), the micromagnetic simulations show a cycloidal magnetic ground state in the absence of an external magnetic field, which is shown in Fig. 5c. Applying a perpendicular magnetic field results in the formation of magnetic Néel skyrmions, which can be distinguished by determining their topological charge (see Supplementary Fig. S39)[2]. Further

increase of the external magnetic field results in a fully polarized magnetic state in agreement with experiment. In addition, we simulated two scenarios in which the anisotropy was allowed to vary across the film. First, we supposed that the anisotropy might have a linear dependence on the distance from the substrate. Second, we supposed that the anisotropy might be stronger close to the interfaces of the sample (see Supplementary Fig. S40). In both cases, the anisotropy was chosen so that, on average, it matches that of the original simulations (see Supplementary Fig. S41). The formation of the skyrmions was not much affected by such variations in anisotropy. Thus, for anisotropy values taken from the current experiment, the profile of the anisotropy within the films does not significantly impact skyrmion formation and stability.

It has previously been reported that DMI fields could be determined in films that have a significant strain gradient via non-reciprocity of surface spin wave propagation[24]. However, this method is not straightforward for our metallic samples with a strain gradient that also has a significant thickness, since the top and bottom interfaces, where the propagating spin waves are located, will also experience different effective magnetic fields, thereby shifting the spin-wave frequency even in the absence of DMI. For this reason, we relied on density functional theory (DFT) calculations to reveal that strain gradients can give rise to DMI values that are large enough to allow for skyrmions to be formed, which we could directly observe using LTEM.

In summary, we have demonstrated the presence of chiral Néel-type spin textures in thin films of simple metallic ferromagnetic alloys that have high Curie temperatures, at elevated temperatures of up to ~773 K. These spin textures arise from the creation of a large strain gradient throughout the thickness of the film via thin-film epitaxy. The strain gradient lowers the symmetry of the centrosymmetric magnetic material and, thereby, gives rise to a DMI within the interior of the film. Our findings significantly broaden the range of potential magnetic materials capable of exhibiting complex chiral spin textures, which are highly relevant for numerous spintronic applications. Thus, our approach to generating high-temperature skyrmions can be extended to other centrosymmetric magnetic systems with high magnetic ordering temperatures.

## Methods

### Thin film growth and characterization

The films were deposited in an AJA "Flagship Series" sputtering system in the presence of Ar gas on $10 \times 10$ mm$^2$ MgO substrates with a [001] orientation. The base pressure before deposition was less than $10^{-8}$ Torr and the Argon pressure during deposition was 3 mTorr. The IrAl, PdAl, RuAl, $Co_x$Al and Co–Ni–Al, alloy thin films were prepared by co-sputtering from individual heavy metal and aluminum targets. The composition of the thin films was calibrated by non-destructive Rutherford Backscattering Spectroscopy (RBS) with an accuracy of ~1–2 atom%. The atomic ratios of IrAl, PdAl, and RuAl were found to be 42:58, 35:65, and 46:54, respectively. The highly resistive TaN-capping layer was prepared by introducing 20% $N_2$ into the Ar gas flow. Magnetization hysteresis loops were measured with a Quantum Design superconducting quantum interference device vibrating sample magnetometer (SQUID-VSM) and for high temperature measurements a Lakeshore VSM.

### X-ray analysis of the strain gradient in thin M$A$l (M = Ru, Pd, and Ir) | Co$_x$Al (Co$_x$Ni$_y$Al) films

The magnitude of the strain gradient, $\nabla \varepsilon_t = \frac{\partial \varepsilon_t}{\partial t}$, along the sample normal ($c$-axis) was analyzed by X-ray diffraction probing the reflection profiles collected by line scans in reciprocal space along the q$_z$ direction in the vicinity of the (002) reflection. The experiments were carried out using a Ga-jet X-ray source ($\lambda = 1.3414$ Å) and a six-circle diffractometer in our laboratory, as well as at the beamline 25b of the

European Synchrotron Radiation Source (ESRF) in Grenoble (France). In both cases, a two-dimensional pixel detector was used to provide high resolution in k-space along the longitudinal scan direction ($q_z$).

Note that the finite instrumental resolution for the Ga-jet X-ray source along $q_z$ was taken into account by using a Gaussian function whose full width at half maximum ($\Delta q_z$) was derived from a scan across the Si (202) reflection ($\Delta q_z$ was found to be $\approx 2.2 \times 10^{-3}$ Å$^{-1}$, which corresponds to $6.5 \times 10^{-3}$ reciprocal lattice units).

Concerning the uniqueness of the model derived from the profile analysis it must be emphasized that in the present study we benefit from several boundary conditions which already define most of the parameters which influence the XRD profile: (i) film thickness, (ii) resolution function (Gaussian function) by which the theoretical curve is convoluted before being compared with the experimental one, and, (iii) the peak position, which is related to a weighted average $c$-lattice parameter. What remains for the fitting procedure is only the vertical strain, $\Delta c/c$, at the top and the bottom of the film in addition to assuming a certain function which interpolates $\Delta c/c$ within the film (e.g., linear, exponential, or sinusoidal profile), although this does not have a big effect on the calculated profiles. In this way, the model which was used in this study is the simplest one consisting of only one independent parameter (the linear gradient). Such strain gradients ($\nabla_t \varepsilon$) tend to asymmetrically broaden the profiles, but only up to a certain point beyond which multiple peaks appear if $\nabla_t \varepsilon$ exceeds a certain limit (which also depends on the film thickness, see Fig. 1d).

### Transmission electron microscopy

For the transmission electron microscopy (TEM) investigations, cross-sectional lamellae from the thin films were prepared by Focused Ion Beam (FIB) Ga+ ion milling using a TESCAN GAIA3 operating at a 30 kV ion-beam energy and standard lift-out procedures. Structural imaging was performed using a JEOL ARM300F2 TEM. For Lorentz TEM imaging several plane-view lamellae were prepared from the as-deposited films, initially by mechanically polishing the backside of the MgO substrate followed by argon ion milling until the high-energy electron beam used in the LTEM experiments can penetrate the lamella. The Ar ion milling was carried out from the back side of the MgO substrate. Magnetic textures were investigated using a FEI TITAN 80-300 TEM and a JEOL JEM-F200 TEM in the Lorentz mode operated at an accelerating voltage of 300 kV and 200 kV, respectively. A Lorentz mini-lens was used for imaging. For high-temperature LTEM experiments, a DENS Lightning biasing & heating double-tilt holder, capable of heating the sample above 900 °C was used. For low-temperature LTEM experiments, we used a GATAN double-tilt LN$_2$ cooling holder (Model 636). This holder allows maximum tilt angles of $\alpha = \pm 29°$ about the $x$ axis and $\beta = \pm 21°$ about the $y$ axis where the $x$ and $y$ axes are in the horizontal plane, and z is along the TEM column. For room-temperature LTEM experiments, a standard JEOL double-tilt holder provided by JEOL company was used. This holder can be tilted about the $x$ and $y$ axes ($\alpha = \pm 36°$ and $\beta = \pm 31°$). A vertical magnetic field was applied to the lamella within the TEM column by passing currents through the coils of the objective lens.

### Freestanding thin film transfer procedure

For the LTEM investigations on samples at various temperatures, a lift-off and transfer method was used. To protect the entire structure, a 100 nm thick PMMA layer was coated onto the as-deposited sample that was prepared by a combination of pulsed laser deposition and sputter deposition in two separate deposition systems (Fig. 4a). The multilayered structure was immersed in deionized water for ~30 min to remove the Sr$_3$Al$_2$O$_6$ layer (Fig. 4a). The separated sheet was then picked up and transferred onto the heating chip. Before the LTEM measurements, the protective PMMA layer was removed by oxygen plasma after drying in nitrogen for 6 h.

## DFT calculations

First-principles calculations were performed using a self-consistent fully relativistic Green function method[54], which is specially designed for semi-infinite systems such as surfaces and interfaces[55]. A generalized gradient approximation was utilized for the exchange-correlation potential[58] and disorder effects were taken into account within a coherent potential approximation as implemented in the multiple scattering theory[56]. The generalized Heisenberg exchange tensor was estimated using the magnetic force theorem[51] adopted for the relativistic case[52,53]. More details about the DFT calculations are given in the supplementary information.

Co$_x$Al alloys ($x > 1.3$) were considered by assuming a structural model in which the (1$a$) site in space group $P4/mmm$ ($x, y, z$) = (0, 0, 0) is completely occupied by Co, while the (1$b$) site ($^1/_2$, $^1/_2$, $^1/_2$) is occupied by both Co and Al atoms. At the experimental composition, Co$_{2.3}$Al, the magnetic moment at the (1$a$) site is -0.5 $\mu_B$, while at the (1$b$) site it is -1.9 $\mu_B$. The Co$_{2.3}$Al films were considered to be semi-infinite on the IrAl substrate. The strain gradient was introduced in a finite part of the Co$_{2.3}$Al films using a model for strain gradient that is mentioned in the main text and supplementary information.

Let us discuss how we determined $A$ and $D$ from our first-principles calculations. The starting point is the atomistic Hamiltonian, which we define in accordance with Zimmermann et al.[59] as,

$$H = -\sum_{ij} J_{ij} e_i \cdot e_j + \sum_{ij} D_{ij} \cdot (e_i \times e_j) + \sum_i K_i (e_i^z)^2 \quad (2)$$

with the direction of the $i$-th magnetic moment denoted as $e_i$. $K_i$, $J_{ij}$, and $D_{ij}$ give the magnitude of the uniaxial anisotropy, the Heisenberg exchange interaction, and the DMI, respectively, all in units of energy. The latter two are accessible by means of the magnetic force theorem[51–53]. Again, similar to Zimmermann et al.[59], we apply the conversion formulas discussed by Schweflinghaus et al.[50], with the minor difference of omitting certain pre-factors. At site $i = 0$, they read

$$\mathscr{A} = \frac{1}{2V} \sum_j J_{0j} \mathbf{R}_j \otimes \mathbf{R}_j \quad (3)$$

$$\mathscr{D} = \frac{1}{V} \sum_j \mathbf{D}_{0j} \otimes \mathbf{R}_j \quad (4)$$

where $\mathbf{R}_j$ is the position of site $j$ with respect to site 0, $V$ is the unit cell's volume and $\otimes$ denotes the tensorial product. Assuming approximate isotropy,

$$A = \frac{1}{3} Tr\{\mathscr{A}\} \quad (5)$$

Following Ma[60], we find that for the case of $C_{4v}$ symmetry, $D$ corresponds to $[D]_{21}$, in other words

$$D = \frac{1}{V} \sum_j D_{0j}^y R_{0j}^x \quad (6)$$

Finally, the average values of $D$ and $A$ are determined for every layer.

## Micromagnetic simulations

We carried out micromagnetic (MM) simulations using the MM solver Mumax3[57]. Some of the material parameters, including the saturation magnetization and the uniaxial magneto-crystalline anisotropy, were determined experimentally. On the other hand, the magnitude of the exchange stiffness $A$ and the Dzyaloshinskii–Moriya interaction (DMI) $D$ were obtained from our first-principles calculations using our in-house Korringa–Kohn–Rostoker Green's Function code Hutsepot[55]. The micromagnetic energy density, $\epsilon = \epsilon(\mathbf{r})$, is defined, following that used in Mumax3, as

$$\epsilon = A(\nabla \mathbf{m})^2 - K(\mathbf{u} \cdot \mathbf{m})^2 + D[m_z(\nabla \cdot \mathbf{m}) - (\mathbf{m} \cdot \nabla)m_z] + \epsilon_{DD} \quad (7)$$

Where $A$ is the exchange stiffness, $K$ is the uniaxial anisotropy constant with regard to the anisotropy axis $\mathbf{u}$, $D$ is the DMI parameter, and $\mathbf{m} = \mathbf{m}(\mathbf{r})$ is the local magnetization direction. The second last expression on the right-hand side accounts for the effects of DMI in the present $C_{4v}$ symmetry[60]. $\epsilon_{DD}$ denotes the contribution from magnetic dipole-dipole interactions. The coordinate system is defined such that the $z$ axis is perpendicular to the film surface.

For the MM simulations, periodic boundary conditions are enforced along the $x$ and $y$ directions, whereas the $z$ direction (perpendicular to the film surface) has a finite size. Mumax3 is based on a finite difference method and thus discretizes space into orthorhombic cells. We choose their size so that twice their side lengths exceed neither the exchange length $l_{ex} = \frac{2A}{\mu_0 M_s^2}$ ($\mu_0$ is the vacuum permeability and $M_s$ is saturation magnetization) nor the domain wall (DW) width $l_{DW} = \sqrt{\frac{A}{K}}$. Furthermore, we assume the material to be invariant under $xy$ translation, but vary $D$ along the $z$-direction according to the first-principles results. $A$ is, to a reasonable approximation, independent of the $z$-direction, and, thus, an average value is assumed for all regions.

Micromagnetic simulations were performed for different magnitudes of the applied magnetic field. The initial state of the sample is set to a state of cell-wise randomly oriented magnetization directions. Next, we minimize the total energy with a sequence of finite temperature Landau–Lifshitz–Gilbert propagations followed by the relax method implemented in Mumax3[57,61,62]. In both cases, the Landau-Lifshitz-Gilbert equation is solved using a Runge-Kutta algorithm.

## Micromagnetic simulations: material parameters

$D = D(z)$ and $A$ are derived from first-principles calculations. In Mumax3, the $z$-variation is reached through the implementation of different regions. Each region's $D$ values are obtained from first-principles results via linear interpolation along the $z$ axis. $M_S$ and $K$ are experimentally determined and, as an approximation, assumed to be independent of $z$ and thus equivalent for all regions.

## Data availability

The data that support the findings of this study are available from the corresponding author upon reasonable request.

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

## Acknowledgements

The authors thank Claudia Münx and Norbert Schammelt for preparing plane-view and cross-sectional TEM lamellae. The authors acknowledge the help of Katayoon Mohseni for the XRD data analysis. We thank Herbert Engelhard for conducting Rutherford Backscattering measurements and Ankit K. Sharma for support in Magnetic Force Microscopy (MFM) imaging. The authors thank the European Synchrotron Radiation Facility (ESRF) for the provision of beamtime. We thank Georg Woltersdorf for very useful discussions. Calculations were carried out at the Rechenzentrum Garching of the Max-Planck Society.

## Author contributions

S.S.P.P. and P.W. initialized the project. P.W. carried out thin film growth, characterization, and magneto-transport measurements. R.S. and H.D. performed in situ LTEM experiments, and R.S. analyzed the LTEM images. H.L.M., J.R.-Z., and E.S.-T. carried out the XRD measurements of the films. K.G. transferred the freestanding films into heating chips and $Si_3N_4$ membranes. A.M. conducted high-temperature VSM measurements. I.K. conducted RBS analysis of the thin film samples. A.E. and D.E. carried out first-principles calculations and micromagnetic simulations. H.D. carried out HRTEM imaging. P.W., R.S., H.L.M., A.E., and S.S.P.P. wrote the original manuscript. P.W., H.L.M., B.P., A.E., and S.S.P.P. wrote the revised manuscript. All authors discussed the data and commented on the manuscript. S.S.P.P. directed and supervised the project.

## Funding

S.S.P.P. acknowledges support from the Deutsche Forschungsgemeinschaft [DFG, German Research Foundation – Project number 403505322 under SPP 2137] and the European Union [ERC Advanced Grant SUPERMINT project number 101054860]. B.P. and S.S.P.P. acknowledge support from the DFG under the German Excellence Strategy [Grant EXC3112/1 533767171, Center for Chiral Electronics]. A.E. discloses support from the Fonds zur Förderung der Wissenschaftlichen Forschung (FWF) [Grant No. I 5384]. Open Access funding enabled and organized by Projekt DEAL.

## Competing interests

The authors declare no competing interests.
