## [Transparent Peer Review file · Nature Communications]

High temperature Néel skyrmions in simple ferromagnets

Corresponding Author: Professor Stuart Parkin

Version 0:

Reviewer comments:

Reviewer #1

(Remarks to the Author)

In their response, the authors have acknowledged the three critical limitations of this work: (1) it motivates futuristic high-temperature applications rather than addressing any BEOL-related skyrmion/MRAM bottlenecks, (2) the vertical strain gradient approach was previously demonstrated on oxides (ref. 27) and here is extended to metallic systems, and (3) unlike ref. 27, they do not present direct experimental DMI evidence. Meanwhile, the authors have also improved clarity on anisotropy determination.

In view of these limitations, Nature Communications is more appropriate for this work. However, for acceptance, the manuscript must still explicitly acknowledge limitations 1 and 3 as per below:

1. p8: The manuscript still conflates "thermal stability" with BEOL "material integrity" requirements, which is misleading. BEOL/CMOS temperature compatibility is routine in MRAM (and skyrmion) materials, and this framing should be removed. The authors should focus on emerging high-temperature electronics (ref. 51), and clearly state these are futuristic applications for which electrical control has to be achieved.
2. In their response, the authors claim that their film thickness (~30 nm) precludes experiments, yet ref. 27 reports direct DMI measurements for ~100 nm thick films. The manuscript must (a) reference ref. 27's direct measurements as establishing the expt standard, and (b) acknowledge the absence of direct DMI measurements, and their reliance on DFT calculations and indirect observations.

With these clarifications, I recommend acceptance for Nature Communications.

REVIEWERS' COMMENTS

Reviewer #1 (Remarks to the Author):

In their response, the authors have acknowledged the three critical limitations of this work: (1) it motivates futuristic high-temperature applications rather than addressing any BEOL-related skyrmion/MRAM bottlenecks, (2) the vertical strain gradient approach was previously demonstrated on oxides (ref. 27) and here is extended to metallic systems, and (3) unlike ref. 27, they do not present direct experimental DMI evidence. Meanwhile, the authors have also improved clarity on anisotropy determination.

In view of these limitations, Nature Communications is more appropriate for this work. However, for acceptance, the manuscript must still explicitly acknowledge limitations 1 and 3 as per below:

1. p8: The manuscript still conflates "thermal stability" with BEOL "material integrity" requirements, which is misleading. BEOL/CMOS temperature compatibility is routine in MRAM (and skyrmion) materials, and this framing should be removed. The authors should focus on emerging high-temperature electronics (ref. 51), and clearly state these are futuristic applications for which electrical control has to be achieved.

2. In their response, the authors claim that their film thickness (~30 nm) precludes experiments, yet ref. 27 reports direct DMI measurements for ~100 nm thick films. The manuscript must (a) reference ref. 27's direct measurements as establishing the expt standard, and (b) acknowledge the absence of direct DMI measurements, and their reliance on DFT calculations and indirect observations.

With these clarifications, I recommend acceptance for Nature Communications.

RESPONSE

We are very pleased that the referee recommends publication in Nature Communications.

- (1) We have added a sentence that electrical control has yet to be demonstrated. And we have clarified the BEOL operation temperature with skyrmions versus the temperature stability of the material itself.
- (2) We have added a paragraph that we infer the presence of DMI from the direct observation of skyrmions using LTEM with support from theoretical calculations of strain gradient induced DMI. With respect to ref. 27 that claims the measurement of DMI from spin wave measurements we are not convinced by the interpretation of these data which we believe is not sound.